# MicroRNAs in Extracellular Vesicles of Alzheimer’s Disease

**DOI:** 10.3390/cells12101378

**Published:** 2023-05-13

**Authors:** Wanran Li, Yun Zheng

**Affiliations:** 1State Key Laboratory of Primate Biomedical Research, Institute of Primate Translational Medicine, Kunming University of Science and Technology, Kunming 650500, China; 2College of Landscape and Horticulture, Yunnan Agricultural University, Kunming 650201, China

**Keywords:** Alzheimer’s disease, microRNA, extracellular vesicles, brain, blood, cerebrospinal fluid

## Abstract

Alzheimer’s disease (AD) is a neurodegenerative disease with dysfunction of memory, language and thinking. More than 55 million people were diagnosed with AD or other dementia around the world in 2020. The pathology of AD is still unclear and there are no applicable therapies for AD. MicroRNAs (miRNAs) play key roles in AD pathology and have great potential for the diagnosis and treatment of AD. Extracellular vesicles (EVs) widely exist in body fluids such as blood and cerebrospinal fluid (CSF) and contain miRNAs that are involved in cell-to-cell communication. We summarized the dysregulated miRNAs in EVs derived from the different body fluids of AD patients, as well as their potential function and application in AD. We also compared these dysregulated miRNAs in EVs to those in the brain tissues of AD patients aiming to provide a comprehensive view of miRNAs in AD. After careful comparisons, we found that miR-125b-5p and miR-132-3p were upregulated and downregulated in several different brain tissues of AD and EVs of AD, respectively, suggesting their value in AD diagnosis based on EV miRNAs. Furthermore, miR-9-5p was dysregulated in EVs and different brain tissues of AD patients and had also been tested as a potential therapy for AD in mice and human cell models, suggesting that miR-9-5p could be used to design new therapies for AD.

## 1. Introduction

As a degenerative neurological disease and the most common form of dementia, Alzheimer’s disease (AD) is marked with dysfunction of memory, language and thinking [1]. Based on the data of the World Health Organization in 2020, 55 million people were diagnosed with AD or other dementia, which was the seventh-largest cause of death [2]. Amyloid plaques and neurofibrillary tangles (NFTs) are the pathologic characteristics of AD and related to the most generally accepted hypotheses, i.e., the amyloid cascade hypothesis and the Tau hypothesis [3]. The amyloid cascade hypothesis posits that the overexpression and dysfunction of APP cause the overproduction of β-amyloid (Aβ) protein, combined with the dysfunction of Aβ clearance resulting in misfolded insoluble protein and amyloid plaque formation, which are the major cause of neuronal dysfunction and death [4,5]. A recent study supplemented this hypothesis. The cellular responses to Aβ in different types of brain cells determined whether the brain’s function could be performed normally [6]. Tau proteins are involved in microtubule formation, which is related to cell development, cell polarity and cellular transport [7]. In the Tau hypothesis of AD, Tau shows hyperphosphorylation and aggregation, leading to the breakdown of microtubule and cellular transport, aggregation of paired helical filaments and degeneration of neurons [8]. Thus, Tau protein is considered to play a key role in AD pathology. Moreover, many studies have suggested that Aβ and Tau are acting collaboratively in AD pathology [9]. Based on their important roles in AD pathology, Tau and Aβ are also important targets in AD treatment [10]. A review summarizing the treatment strategies for AD indicated that therapies targeting Aβ did not yield the results expected in clinical trials [11]. However, therapies targeting Tau are in the early stage of development and have a great therapeutic potential in the future [11].

In addition to the amyloid cascade and Tau hypotheses of AD, three other hypotheses have been reported. Firstly, ApoE protein, an apolipoprotein, is thought to be a strong genetic risk factor for late-onset AD (LOAD) [12]. A longitudinal study of individuals from families with autosomal dominant AD suggested that the ApoE allele may promote amyloid deposition [13]. ApoE4, a variant protein product of the APOE gene, induced more Tau hyperphosphorylation and GABAergic neuron degeneration compared to ApoE3 [14]. In mice, the expression of ApoE4 also leads to the blood–brain barrier breakdown, promoting neuronal dysfunction [15]. Secondly, Efthymiou and Goate [16] propose that several genes, i.e., CR1, SPI1, the MS4As, TREM2, ABCA7, CD33 and INPP5D, are expressed by microglia and contribute to AD in a non-Aβ-dependent fashion. In particular, Gratuze et al. [17] summarized the role of TREM2 in AD development. TREM2 is involved in the innate immune system and affects Tau and amyloid pathologies [17]. Based on the findings of the dysregulated immune response in AD development, a novel model for amyloidogenesis, the antimicrobial protection hypothesis, was proposed. Aβ deposition is thought to be an early immune response in this hypothesis. However, the high level of Aβ deposition induced microglial activation, which, combined with amyloid plaques and neurofibrillary tangle pathology, mediated neurodegeneration [18]. Thirdly, several studies uncovered that sleep impairment could play an important role in AD development. A study of isotope tracing Aβ peptides in human cerebrospinal fluid (CSF) showed increased Aβ production in sleep-deprived participants [19]. Another study showed a similar result for Tau production in both humans and mice [20].

MiRNAs are small non-coding RNAs, usually with 22 nucleotides [21]. MiRNAs are transcribed in the nucleus, and their primary transcripts are called pri-miRNAs [22]. Pri-miRNA is processed into a hairpin structure through the RNase III enzyme Drosha and other cofactors and then processed into a shorter hairpin, called pre-miRNA (precursor miRNA), under the participation of several proteins [23,24]. Then pre-miRNA is transported to the cytoplasm and the loop is cleaved to produce a double-stranded RNA called an miRNA:miRNA* duplex [21]. One strand of the miRNA:miRNA* duplex is loaded with AGO protein to form the silencing complex. Meanwhile, the other strand is discarded [25,26,27,28]. MiRNAs usually suppress the expression of their targets by complementary binding to the target mRNAs [29,30]. MiRNAs play key roles in many biological or pathological processes, including metabolism, proliferation, differentiation, development, apoptotic cell death, viral infection and molecular mechanisms of diseases [29].

Extracellular vesicles (EVs) are membrane-enclosed vesicles secreted by cells [31,32]. In consideration of differences in their size, biogenesis, cellular origin and biophysical properties, EVs can be divided into exosomes, microvesicles and apoptotic bodies [33]. Exosomes (30–100 nm in diameter) and microvesicles (100 nm to 1000 nm in diameter) are released from healthy living cells [34], while apoptotic bodies (800–5000 nm in diameter) are produced during programmed cell death [35]. A study of distinct RNA in extracellular vesicles from three different kinds of cell lines including a human erythroleukemia cell line (TF-1), a human mast cell line (HMC-1) and a mouse microglia cell line (BV-2) indicated that there was a little or no RNA in microvesicles except for those collected from TF-1 cells [35]. Meanwhile, Budnik et al. [34] thought that apoptotic bodies might not be involved in transcellular communication in the nervous system, since they were engulfed by phagocytic cells rapidly after being released. Only exosomes are involved in cell-to-cell communication in the central nervous system [36]. Exosomes from neurons regulate the integrity of the brain vascular system [37], and exosomes from astrocytes may impact neuronal morphology and function [38].

EVs contain different kinds of molecules, such as nucleic acids and proteins [39]. The miRNAs of exosomes keep similarities with the parent cells in cancer [40] and the proteins of EVs consist of membrane proteins and the lumen of EVs [39]. It is reasonable to expect that pathological miRNAs in EVs might be detected earlier and be more notable than those of proteins in EVs. Indeed, it has been reported that miRNAs in exosomes are dysregulated in AD patients when compared to healthy people [36,41]. To provide a comprehensive view of miRNAs in EVs of AD and their relations to AD, this review summarizes the miRNAs of extracellular vesicles in different samples of AD patients and their potential roles in AD pathology, diagnosis and treatment.

## 2. Relevant miRNAs in AD

Several studies revealed that miRNAs were involved in AD pathology, usually through targeting AD-related genes or signalling pathways, as summarized in Table 1. As a brain-rich miRNA, miR-9-5p has been investigated in many studies. As listed in Table 1, miR-9-5p targets several AD-related genes including BACE1, SIRT1, CAMKK2 [42] and TGFBIp [43]. Aβ is formed during the cleavage of the amyloid precursor protein (APP) by BACE1 [44]. CAMKK2 has been inhibited by miR-9-5p in AD resulting in the aggravation of Aβ-induced synaptotoxic impairment [45]. Interestingly, SIRT1 was studied as a protective factor in AD by inhibiting the accumulation of Aβ and neuroinflammation [46]. Although the level of miR-9-5p was downregulated in AD brains and EVs from CSF [47,48,49], the expression of SIRT1 was decreased in AD patients [46]. This suggests that the downregulated miR-9-5p might be a kind of positive response in AD development, but the other regulators suppressed the production of SIRT1, except for miR-9-5p. Interestingly, Sethi and Lukiw found that miR-9-5p was upregulated in the temporal lobe neocortex of AD brains [50]. The mutation of TGFBIp accelerated Aβ aggregation [51]. All these targets of miR-9-5p are related to the Aβ accumulation in AD. Furthermore, miR-9-5p also decreased the Tau degradation by targeting UBE4B [52].

In addition to the contribution to the Aβ accumulation and Tau hyperphosphorylation, miR-9-5p is also related to apoptosis [53]. Zhang et al. [54] reported that apoptosis might be associated with neuron loss in AD since Aβ could induce neuronal apoptosis. In the forebrain, miR-9-5p regulates the expression of hairy1, which mediates neurogenesis, apoptosis and proliferation [53] (Table 1). Moreover, miR-9-5p was downregulated in AD cell models, which resulted in an increased expression of GSK-3β[55]. Moreover, the overexpression of GSK-3β contributed to cell apoptosis, oxidative stress and mitochondrial dysfunction [55]. Therefore, miR-9-5p is also involved in AD development through other ways, such as apoptosis and oxidative stress.

Other miRNAs play important roles in AD pathology too. As listed in Table 1, these miRNAs can be generally divided into two categories. The first category includes miRNAs related to Aβ accumulation. For example, miR-17-5p, miR-20a and miR-106b repressed the expression of APP [56] (as listed in Table 1). miR-153 [57] and miR-193b [58] also inhibited the level of APP (as listed in Table 1). miR-124 was involved in the splicing of APP mRNA [59] (see Table 1). miR-144 and miR-451 were reported as inhibitors of ADAM10 [60] (as listed in Table 1). The ADAM10 protein inhibits the generation of Aβ[60]. miR-107 was downregulated in the brains of AD patients compared to the HC and played a positive role in the generation of Aβ by decreasing the inhibition of BACE1 [61] (as listed in Table 1). Similarly, miR-9-5p, miR-29a, miR-29b-1 [47], miR-29c [62], miR-339-5p [63] and miR-485-5p [64] are also BACE1 inhibitors (as listed in Table 1).

The second category includes miRNAs involved in Tau hyperphosphorylation (see Table 1). For example, miR-15a might be a potential regulator of ERK1 [65]. miR-125b-5p upregulated the expression of Erk1/2, an activator of kinase cdk5, and was involved in Tau phosphorylation in AD development [66]. miR-125b-5p also decreases the production of phosphatases, including DUSP6 and PPP1CA [66] (see Table 1). miR-26b was up-regulated in AD and might induce cell apoptosis, aberrant cell cycle entry and increasing Tau-phosphorylation [67] (see Table 1). It was reported that miR-219 directly regulated the synthesis of Tau protein by binding the 3’-UTR of the Tau mRNA [68] (see Table 1).

**Table 1 cells-12-01378-t001:** The miRNAs with relevant functions in AD.

MicroRNAs	Pathological Roles	Reference
let-7e-5p	Let-7e in small neuron-derived extracellular vesicles can trigger inflammatory responses in microglia by increasing inflammatory cytokines, such as interleukin 6.	[69]
miR-9-5p	Accompanied by STUB1, decreases the clearance of Tau proteins by targeting UBE4B.	[52]
	Alleviated Aβ-induced synaptotoxic impairment by inhibiting CAMKK2.	[45]
	Related to the amyloid cascade hypothesis and memory loss by targeting BACE1, SIRT1 and CAMKK2.	[42]
	Repressed BACE1 and TGFBIp mRNA.	[43]
		[47]
	Mediated neurogenesis, apoptosis and proliferation by regulating the expression of hairy1.	[53]
	Induced oxidative stress, mitochondrial dysfunction and cell apoptosis through targeting GSK-3β.	[55]
	As the target of miR-9-5p, SIRT1 suppressed Aβ accumulation and attenuated neuroinflammation.	[46]
miR-15a	Might be a potential regulateor of ERK1, which is a candidate of kinase in Tau phosphorylation.	[65]
miR-17-5p, miR-20a, miR-106b	Repressed APP expression in neuronal cell lines.	[56]
miR-26b	Might induce aberrant cell cycle entry, increasing Tau-phosphorylation and cell apoptosis.	[67]
miR-29a, miR-29b-1	Repressed BACE1 expression in vitro.	[47]
miR-29c	Repressed the expression of BACE1.	[62]
miR-34a	Related to the energy metabolism, synaptic plasticity and resting-state network activity by targeting several genes.	[70]
miR-107	Contributed to the Aβ generation by targeting BACE1.	[61]
miR-124	Involved in APP mRNA alternative splicing.	[59]
miR-125b-5p	Might play a pro-inflammatory role.	[71]
	Contributed to Tau hyperphosphorylation by upregulating the expression and activity of kinases, such as Erk1/2 and p35, an activator of kinase cdk5, and downregulating phosphatase production, including DUSP6 and PPP1CA.	[66]
miR-132-3p, miR-212	Inhibited apoptosis by regulating PTEN, FOXO3a and P300 and against oxidative stress.	[72]
miR-146a-5p	Contributed to the inflammatory response in AD by targeting CFH.	[73]
	Mediated downregulation of the IRAK-1. It combined with NF-κB induced upregulation of IRAK-2, resulting in the inflammatory response.	[74]
miR-153	Mediated the miRNA-induced suppression of APP.	[57]
miR-193b	Repressed the expression of APP.	[58]
miR-219	Directly regulated the production of Tau protein by mediating the miRNA-induced suppression of Tau protein.	[68]
miR-339-5p	Repressed the expression of BACE1.	[63]
miR-144, miR-451	Repressed the expression of ADAM10.	[60]
miR-485-5p	Mediated the miRNA-induced suppression of BACE1.	[64]

Some miRNAs have other contributions to AD development. As listed in Table 1, miR-125b-5p (miR-125b) was upregulated in the CSF of AD [71]. miR-125b-5p was also specifically over-expressed in microglia compared to other immune cell types, suggesting its role in the innate immune response [71]. Similar to miR-125b-5p, let-7e-5p also triggered an increase in pro-inflammatory cytokines (such as interleukin 6) and inflammation in microglia [69]. miR-146a-5p was upregulated in AD brains and contributed to the inflammatory pathology in AD by targeting complement factor H (CFH) [73] and IRAK-1 [74] (see Table 1). The target genes of miR-34a were related to energy metabolism, synaptic plasticity and resting-state network activity [70], suggesting that miR-34a might be an important regulator in AD development (see Table 1). miR-132-3p (miR-132) and miR-212 inhibited apoptosis by regulating the expression of PTEN, FOXO3a and P300 and acted against oxidative stress in vitro [72] (as listed in Table 1).

## 3. Dysregulated miRNAs in Brains of AD Patients

In order to know whether there are common dysregulated miRNAs in the EVs and brain tissues of AD patients, we summarized the dysregulated miRNAs in different brain regions of AD patients in Table 2. In a study of the entorhinal cortex (EC) and superior temporal gyrus (STG), three miRNAs (miR-129-5p, miR-132-5p, miR-138-5p) were downregulated and miR-195-5p was upregulated in 99 AD brain samples compared to 91 healthy controls (HC) [75] (see Table 2). In another study of an unknown brain region, three upregulated miRNAs and 13 downregulated miRNAs were found in five AD brains compared to the five normal controls [47] (as listed in Table 2).

The Braak stage, also called the Tau Braak stage, reflects the changes in Tau hyperphosphorylation in AD development. It can be divided into six stages [76]. The Braak stage is usually used for the stage classification of AD with different criteria in different studies [48,61,63,66,67,70,77]. Since the disease changes happening in Braak III correspond to the early symptoms of AD and are defined as mild cognitive impairment (MCI) [78], some studies would regard the Braak stages I and II as a control group [48,66,67,70,77]. The Braak stages IV–VI are classified as a late stage or severe stage of AD [48,66,67,70,77]. The Braak stage was often used in AD studies and we summarized the information about the Braak stage in Table 2 when available.

miR-29a was downregulated in late-stage AD patients (Braak IV or VI) [77] (as listed in Table 2). miR-125b-5p was upregulated in AD (Braak V and VI) brains and miR-29a and miR-29b were downregulated compared to HC (Braak I and II) [66]. miR-26b was upregulated in both very early AD (Braak III) and severe AD (Braak VI) patients [67] (see Table 2). miR-34a and miR-146a-5p were upregulated in the temporal cortex of both mild (Braak III) and severe AD (Braak VI) patients [70] (as listed in Table 2). In a study by Cogswell et al. [48], Braak I was regarded as the normal control, and Braak III–VI were regarded as AD. They did not conduct the early- and late-stage classification of AD. In this study, 12 miRNAs were upregulated and six miRNAs were downregulated in AD brains [48] (see Table 2). Different from the previous studies, Long et al. took Braak I–III as the control group [63]. They found that miR-339-5p was downregulated in AD brains when compared to HC [63] (see Table 2). Wang et al. [61] separated all cases into four groups in their research, including nondemented with no/negligible AD-type pathology, nondemented with incipient AD pathology, MCI and AD. Among the four groups, miR-107 was downregulated in AD and MCI compared to nondemented with no/negligible AD-type pathology [61] (see Table 2).

We also found that the Braak stage combined with other pathological features were used to classify the stages of AD. As listed in Table 2, three dysregulated miRNAs (one upregulated miRNA and two downregulated miRNAs) were found in a 20-individual study with clear criteria of samples [79]. The authors combined the neurofibrillary tangles and neuritic plaques with clinical cognition evaluations to categorize the AD and HC groups [79]. Lau et al. [80] found 35 miRNAs dysregulated in the hippocampus (HP) and 41 miRNAs in the prefrontal cortex (PC) (see Table 2). For the HP cohort, Lau et al. [80] gave detailed information of the involved individuals but did not have a clear criterion for AD patients and normal controls. For the PC cohort, Lau et al. [80] found that 41 miRNAs were expressed differently in different Braak stages including the controls (Braak 0), early stages (Braak I and II), mid stages (Braak III and IV) and late stages (Braak V and VI). Santa-Maria et al. [68] found that miR-219 was downregulated in AD (Braak V and VI) and severe primary age-related tauopathy (Braak III and IV) compared to HC (see Table 2). Some studies did not use the Tau Braak stages in AD description. Lukiw et al. [73] used the criteria of the center to establish a registry for Alzheimer’s disease/National Institutes of Health to distinguish the AD group and HC group (as listed in Table 2). They found that miR-146a-5p was upregulated in AD brains when compared to HC [73].

**Table 2 cells-12-01378-t002:** Dysregulated miRNAs in brain tissues of AD patients.

Sample (AD/Control)	Brain Region ${}^{*}$	Upregulated miRNAs	Downregulated miRNAs	Braak Stage	Reference
190 (99/91)	EC and STG	miR-195-5p	miR-129-5p, miR-132-5p, miR-138-5p	/	[75]
10 (5/5)	/	miR-197, miR-320, miR-511	let-7i, miR-9-5p, miR-15a, miR-19b, miR-22, miR-26b, miR-29a, miR-29b-1, miR-93, miR-101, miR-106b, miR-181c, miR-210, miR-363	/	[47]
11 (7/4)	FC	/	miR-29a	Braak IV or VI	[77]
15 (10/5)	FC	miR-125b-5p	miR-29a, miR-29b	Braak V and VI	[66]
28 (20/8)	TC	miR-26b	/	Braak III and VI	[67]
19 (10/9)	TC	miR-34a, miR-146a-5p	/	Braak III and VI	[70]
27 (20/7)	HP; MFG; CB	miR-27a, miR-27b, miR-30e-5p, miR-34a, miR-92, miR-100, miR-125b-5p, miR-145, miR-148a, miR-381, miR-422a, miR-423	miR-9-5p, miR-210, miR-212, miR-132-3p, miR-146b, miR-425	Braak III-VI	[48]
25 (20/5)	FC	/	miR-339-5p	Braak IV-VI	[63]
22 (12/11)	TC	/	miR-107	Braak III-V	[61]
20 (11/9)	FC	miR-32-5p	miR-182-5p, miR-1304-5p	/	[79]
64 (41/23)	HP	let-7f-5p, let-7i-5p, miR-23a-3p, miR-27a-3p, miR-92b-3p, miR-142-3p, miR-150-5p, miR-195-5p, miR-199a-3p, miR-199b-3p, miR-200a-3p, miR-223-3p, miR-362-3p, miR-363-3p, miR-455-5p	miR-124-3p, miR-127-3p, miR-128, miR-129-2-3p, miR-129-5p, miR-132-3p, miR-136-5p, miR-138-5p, miR-219-2-3p, miR-329, miR-370, miR-409-5p, miR-410, miR-425-5p, miR-433, miR-487a, miR-487b, miR-495-3p, miR-543, miR-769-5p	/	[80]
41 (34/7)	PC	miR-27a-3p, miR-92b-3p, miR-190b, miR-200a-3p, miR-214-3p, miR-424-5p, miR-517c-3p, miR-519a-3p, miR-744-5p, miR-874, miR-1260a, miR-1275	miR-127-5p, miR-129-2-3p, miR-10b-5p, miR-129-5p, miR-132-3p, miR-133b, miR-135b-5p, miR-136-5p, miR-210, miR-219-1-3p, miR-337-3p, miR-370, miR-382-5p, miR-409-5p, miR-421, miR-431-5p, miR-485-5p, miR-487a, miR-491-3p, miR-496, miR-508-3p, miR-520a-3p, miR-548j, miR-551b-3p, miR-633, miR-758-3p, miR-1178-3p, miR-1179, miR-1321	Braak I-VI	
27 (7/20)	/	/	miR-219	Braak I-VI	[68]
46 (23/23)	HP and TL	miR-146a-5p	/	/	[73]
22 (11/11)	ATC	/	miR-124, miR-107	/	[59]
66 (36/30)	STLN	miR-146a-5p	/	/	[74]
42 (21/21)	HP and CB	miR-132-3p, miR-152	miR-138, miR-139-5p, miR-149, miR-191, miR-204, miR-328, miR-370	/	[81]
8 (4/4)	PLC	miR-134, miR-185, miR-188, miR-320, miR-382, miR-432, miR-486, miR-572, miR-575, miR-601, miR-617, miR-671, miR-765	miR-15a, miR-20b, miR-29b, miR-29c, miR-30e-5p, miR-95, miR-101, miR-130a, miR-148b, miR-181c, miR-368, miR-374, miR-376a, miR-494, miR-582, miR-598	/	[82]
18 (8/10)	/	/	miR-15a	/	[65]
60 (31/29)	/	/	miR-29c	/	[62]
30 (19/11)	ATC or CB	/	miR-106b	/	[56]
46 (30/16)	TC	/	miR-132-3p, miR-212	/	[72]
21 (10/11)	HP	miR-16, miR-34c, miR-146a-5p	/	Braak III and IV	[83]
21 (10/11)	HP	/	miR-16, miR-107, miR-128a, miR-146a-5p	Braak VI	
20 (10/10)	FC	/	miR-153	Braak III-VI	[57]
10–24 (5–12/5–12)	PL, CB, HP, EC	/	miR-485-5p	/	[64]
12 (6/6)	TLN	miR-9-5p, miR-125b-5p, miR-146a-5p	/	/	[50]
16 (8/8)	STG and MTG	/	miR-100, miR-132-3p	/	[84]

* The abbreviations of brain regions are ATC: anterior temporal cortex; CB: cerebellum; EC: entorhinal cortex; FC: frontal cortex; HP: hippocampus; MTG: middle temporal gyrus; PL: parietal lobe; STG: superior temporal gyrus; STLN: superior temporal lobe neocortex; TL: temporal lobe; and TLN: temporal lobe neocortex.

Smith et al. [59] found that miR-124 and miR-107 were downregulated in AD brains with an unclear description of AD group (see Table 2). Cui et al. [74] found that miR-146a-5p was upregulated in AD brains (as listed in Table 2). A 42-brain-sample study found two upregulated miRNAs and seven downregulated miRNAs in AD compared to HC [81] (see Table 2). Although the Braak stage information was contained in the description of patients, a clear criterion for the AD and HC groups was not given in this study [81]. Some studies focused more on the function of dysregulated miRNAs and ignored the specific description of AD individuals. For example, Lei et al. [62] found that miR-29c was downregulated in AD brains (see Table 2). Nunez-Iglesias et al. [82] found 29 dysregulated miRNAs from the parietal lobe cortex (PLC) of AD patients. They focused more on the relationship of mRNA and miRNA [82] (see Table 2). miR-15a [65], miR-29c [62], miR-106b [56], miR-132-3p and miR-212 [72] were downregulated in AD brains and the authors paid more attention to the function of dysregulated miRNAs as well (as listed in Table 2).

The expression levels of miRNAs were altered not only at different stages of AD development but also in different brain regions. miR-16, miR-34c and miR-146a-5p were upregulated in early AD (Braak III and IV), while miR-16, miR-107, miR-128a and miR-146a-5p were downregulated in late AD (Braak VI) compared to HC (Braak I and II) [83] (see Table 2). The expression level of miR-153 did not have a significant difference between HC and Braak I and II, Braak III and IV or Braak V and VI [57] (see Table 2). However, miR-153 was significantly downregulated in Braak III–VI compared with HC and Braak I and II [57]. A similar condition also happened in a different brain region. Faghihi et al. [64] found that miR-485-5p was significantly downregulated in the parietal lobe, cerebellum, entorhinal cortex and hippocampus (as listed in Table 2), but was not significantly changed in the cerebellum and superior frontal gyrus.

Some studies compared dysregulated miRNAs in AD to those in other neurogenic diseases [50,84]. For instance, miR-9-5p, miR-125b-5p and miR-146a-5p were upregulated in AD brains, but not in other neurogenic diseases, such as amyotrophic lateral sclerosis, Parkinson’s disease and schizophrenia [50] (as listed in Table 2). Similarly, miR-100 was only downregulated in AD brains compared with HC, but had no statistical changes in frontotemporal lobar dementia (FTLD) and progressive supranuclear palsy (PSP) [84] (see Table 2). However, miR-132-3p was downregulated in AD, FTLD and PSP [84].

## 4. MiRNAs in EVs of Different Biological Fluids of AD Patients

### 4.1. Dysregulated miRNAs in Blood EVs

The miRNAs in EVs of blood samples from AD patients will be discussed first. There are three types of blood samples, comprising plasma, serum and whole blood. As a very common and easy-to-collect body fluid, plasma appears in many AD-related studies. Many miRNAs of EVs isolated from plasma are dysregulated in AD patients [79,85,86,87,88] (as listed in Table 3). A study of exosomes in 70 plasma samples (35 AD and 35 HC) showed that four miRNAs were upregulated, whereas 16 miNRAs were downregulated in AD compared to HC [85] (see Table 3). In another study, six miRNAs were downregulated in EVs derived from the plasma of AD patients compared with HC, but no miRNA was statistically upregulated in the same study [86] (see Table 3). In another study, 37 dysregulated miRNAs (15 upregulated miRNAs and 22 downregulated miRNAs) were found in the plasma EVs of AD patients compared with HC [87] (as listed in Table 3). miR-132-3p and miR-212-3p were downregulated in neural exosomes derived from the plasma of AD patients [79] (see Table 3). Serpente et al. also found several dysregulated miRNAs (two upregulated miRNAs and one downregulated miRNA) in neural EVs collected from the plasma of AD patients [88] (see Table 3). Durur et al. [69] found that let-7e-5p was upregulated in small neuron-derived EVs (sNDEV) of 23 AD patients’ plasma compared to 28 HC. The last three studies used neural EVs, which are not mentioned in other studies. Cheng et al. [89] indicated that the expression of miRNAs in EVs from peripheral blood had a lower correlation with that of the EVs from brain tissues because other organs also released EVs to the blood. The isolation method of brain-derived EVs in the blood should be emphasized to find more AD-related miRNAs [89].

As another common peripheral biofluid, serum is also used frequently in AD studies [49,58,90,91,92,93]. A study found that 17 miRNAs were upregulated and three miRNAs were downregulated in the serum sample of 53 AD compared to 62 controls [49] (see Table 3). Another study found that 14 miRNAs were upregulated and three miRNAs were downregulated in the serum sample of 31 AD compared to 28 controls [90] (as listed in Table 3). miR-135a-5p and miR-384 were upregulated and miR-193b was downregulated in the serum of 208 AD compared to 228 controls [91] (see Table 3). However, Liu et al. found that miR-193b was upregulated in ABCA1-labeled exosomes of AD patients’ serum [92] (see Table 3). The reason for the opposite change of miR-193b from the last two studies might be the different methods of exosome isolation applied. Liu et al. [92] found that an exosome protein, ATP-binding cassette transporter A1 (ABCA1), was significantly upregulated in the CSF samples of AD patients than in those of controls. In [91], all exosomes in the serum of peripheral blood were analyzed, and ABCA1-labeled exosomes were captured and analyzed in [92].

Liu et al. found that ABCA1-labeled exosomal miR-135a-5p was upregulated in the serum of MCI and AD as well [94] (as listed Table 3). Dong et al. found 24 dysregulated miRNAs in the RNA sequence profile and validated five of them with quantitative reverse transcription PCR (qRT-PCR) [93] (see Table 3). Three out of five had significant differences but all five miRNAs had same tendency in RNA sequencing and qRT-PCR [93]. Similar to the results of Dong et al. [93], miR-193b was downregulated in AD and verified with quantitative PCR (qPCR) [58] (see Table 3).

The last type of blood sample is whole blood without any extra treatment after collection. Several dysregulated miRNAs were found in the whole-blood sample of AD patients [31] (see Table 3). miR-146a-5p was downregulated in EVs from the whole blood of nine moderate AD and 13 severe AD compared with 19 controls [31] (as listed in Table 3). Furthermore, let-7g and miR-142-3p were downregulated in EVs from the whole blood of nine moderate AD only compared to the 19 controls [31]. No significantly upregulated miRNAs were found in the EVs of AD patients’ whole blood [31].

### 4.2. Dysregulated miRNAs in Cerebrospinal Fluid EVs

The cerebrospinal fluid (CSF) contains many materials that are important to the brain function. Combined with the choroid plexus, the CSF forms the blood–CSF barrier (B-CSF-B). The B-CSF-B and the blood–brain barrier (BBB) form the barrier of the central nervous system, preventing substances in the blood from reaching the brain directly [95]. AD is divided into early-onset Alzheimer’s disease (EOAD) and late-onset Alzheimer’s disease (LOAD) [96]. A study about the miRNA expression of exosomes isolated from the CSF of EOAD patients found that miR-125b-5p was upregulated and miR-16-5p, miR-451a and miR-605-5p were downregulated compared to HC (see Table 4) [36]. The study also found the same altered tendency in LOAD patients and HC, except for miR-16-5p [36]. There is no significant difference in miR-16-5p expression between LOAD and HC [36], suggesting that the three common miRNAs might be potential candidates to distinguish AD from HC. Another study showed more dysregulated miRNAs in cell-free CSF [49] (see Table 4). Forty-one miRNAs were downregulated in AD [49]. Moreover, miR-193b was upregulated in ABCA1-labeled exosomes derived from the CSF in different stages of AD, including subjective cognitive decline, MCI and AD [92] (as listed in Table 4). The same condition happened in miR-135a-5p too [94] (see Table 4). miR-135a-5p in ABCA1-labeled exosomes derived from the CSF was also upregulated in 32 AD compared to seven controls [94]. Different from the ABCA1-labeled exosome, miR193b in exosomes isolated from the CSF was downregulated in AD [58] (as listed in Table 4).

## 5. Application of EV miRNAs in Diagnosis of AD

To figure out the possibility of exosomal miRNAs as AD biomarkers, McKeever et al. [36] performed logistic regression for the significantly dysregulated miRNAs in exosomes of AD patients’ CSF. According to ROC analysis, miR-451a has the best performance in distinguishing EOAD or LOAD from HC (AUC = 0.95 and 0.85, respectively, as listed in Table 5) [36]. Another study of exosomes in plasma obtained a similar performance using three different machine learning algorithms on seven miRNAs with AUC values of 0.83 to 0.92 [85] (see Table 5). miR-132-3p (AUC = 0.77) and miR-212-3p (AUC = 0.84) from exosomes in plasma were used as biomarkers to distinguish AD from HC, respectively [79] (as listed in Table 5). Durur et al. used let-7e-5p in sNDEV derived from the plasma to distinguish AD patients from HC, and the AUC was 0.92 [69]. miR-135a-5p, miR193b and miR-384 were used to distinguish AD from HC with a high sensitivity and specificity [91] (see Table 5). These three miRNAs were also used to distinguish MCI from HC separately and in combination [91]. The highest AUC (0.995) was obtained when the combination of miR-135a-5p, miR193b and miR-384 was used in the diagnosis of MCI [91]. Similarly, miR-22-3p, miR-30b-5p and miR-378a-3p were used to build a model for AD prediction as well [93] (see Table 5). However, the number of samples was smaller than that in Yang et al. [91] and the performance of the model was much worse than those in Yang et al. [91].

In addition to the machine learning methods, potential diagnostic methods based on cell conversion and differentiation have appeared [97,98]. Drouin-Ouellet et al. [97] successfully converted human dermal fibroblasts from patients with AD, Parkinson’s disease and Huntington’s disease into neurons by inhibiting RE1-silencing transcription factor expression. The conversion was partially regulated by miR-9-5p and miR-124 [97]. Ishikawa et al. [98] also achieved a rapid differentiation of neurons from human pluripotent stem cells (iPSCs) through the Neurogenin2 gene, miR-9-5p and miR-124. These neurons converted from iPSCs could be used for disease diagnosis, drug screening and in disease model studies [98].

## 6. Application of miRNAs in AD Treatment

Some miRNAs have been tested as therapeutic targets of AD in vitro or in animal model experiments (as listed in Table 6). For example, osthole, a natural coumarin derivative, has protective effects in neurons of both mice and humans by preventing cell death, improving cell viability and increasing the production of synaptic proteins [99] (see Table 6). Osthole decreases the expression of CAMKK2 and p-AMPKα through upregulating miR-9-5p to achieve the therapeutic effects [99]. Osthole also stimulates the differentiation of neural stem cells into neurons by upregulating the expression of miR-9-5p, which inhibits the Notch signaling pathway [100]. miR-16-5p mimics induced a reduction in AD-related genes, including APP, BACE1 and Tau, in AD mice [101] (as listed in Table 6). The overexpression of miR-16-5p decreased the apoptosis of neurons in a cellular AD model [102] (see Table 6). miR-124 was downregulated in AD patients [103] (as listed in Table 6). Delivering miR-124 and rutin, a small molecular ancillary drug, to the brain of AD mice significantly increased miR-124 and inhibited the expression of BACE1 and APP [103].

Based on the biological functions of miRNAs in gene regulation and the roles EVs play in cell communication [106], miRNAs were delivered using EVs to achieve the therapeutic purpose. MiRNAs or short interfering RNA (siRNA) can be delivered to inhibit the expression of pathological genes in AD. For example, siRNA can be delivered to neural cells specifically with exosomes, resulting BACE1 gene knockdown in AD mice [107]. miR-29b was contained in the exosomes and its target genes included BACE1 and BIM [104] (Table 6). The injection of miR-29b-containing exosomes reduced the pathological effects of Aβ in a rat model of AD [104].

For the miRNAs to play a positive role in AD pathology, the miRNAs inhibitors must have therapeutic effects. For example, the EVs of astrocytes treated with aFGF or miR-206-3p inhibitor reduced the expression of miR-206-3p, improved synaptic plasticity and cognitive deficits and attenuated Aβ accumulation in AD mice [105]. The repression of miRNAs that are upregulated in the EVs of AD patients can yield therapeutic effects.

## 7. Discussion

We first summarized the dysregulated miRNAs to find the miRNAs appearing in different brain regions. As shown in Figure 1A and Table A1, eight upregulated miRNAs and 16 downregulated miRNAs were found in at least two different brain regions. Among these miRNAs, miR-146a-5p [50,70,73,74,83] and miR-132-3p [48,72,80,81,84] were reported to be dysregulated in six independent AD studies, respectively. miR-146a-5p is related to the inflammatory response in AD [73,74]. Combined with miR-212, miR-132-3p is related to apoptosis [72]. Interestingly, miR-146a-5p was upregulated in the brains of AD patients compared to HC [50,70,73,74,83], but one study reported that miR-146a-5p was downregulated in the brain of AD patients compared to HC [83]. Similar to miR-146a-5p, miR-132-3p was reported to be downregulated in AD in five independent studies [48,72,80,84] but up-regulated in AD in only one independent study [81]. More studies are needed to verified the dysregulation of miRNAs in AD.

To determine whether there are common miRNAs with the same tendency of EVs derived from different body fluids, the upregulated and downregulated miRNAs are summarized in Figure 1B,C respectively. As shown in Figure 1B, no upregulated miRNAs were found in EVs derived from whole blood. Twenty-two and 52 upregulated miRNAs were found in EVs derived from plasma and serum, respectively. Only miR-22-3p was in common. As a dysregulated miRNA, miR-22-3p was used as an AD biomarker to distinguish AD from HC [93]. miR-125b-5p, miR-135a-5p and miR-193b were common in EVs in serum and CSF, suggesting the exchange between the CSF and peripheral blood. miR-125-5p [85], miR-135a-5p and miR-193b [91] were used in AD identification too. miR-125-5p might be a positive factor in AD development by inducing Tau hyperphosphorylation [66] and being specifically expressed in microglia [71].

The downregulated miRNAs in EVs from different body fluids are shown in Figure 1C. Forty-seven, 11, three and 45 downregulated miRNAs in EVs were found from plasma, serum, whole blood and CSF, respectively. miR-21-5p, miR-342-3p and miR-375 were common in EVs of plasma and serum. miR-132-3p, miR-139-5p and miR-451a were common in EVs of plasma and CSF. miR-124-3p and miR-193b were common in EVs of serum and CSF. Interestingly, one common miRNA, miR-193b, was downregulated in exosomes from serum [58,91] and CSF [58], but upregulated in ABCA1-labeled exosomes from serum and CSF [92], indicating that the different methods of exosome isolation might influence the results. As shown in Figure 1D, four upregulated miRNAs and eight downregulated miRNAs were found in EVs of at least two different types of body fluid samples (details shown in Table A2).

To answer whether body fluids carry the deregulation patterns of miRNAs in the brain tissues of AD patients, we compared the upregulated miRNAs in EVs and brain tissues of AD patients (Figure 1E). Seventy-three and 59 upregulated miRNAs were found in EVs and brain tissues, respectively. Eight common miRNAs were found in both brain tissue and EVs of plasma or serum. They are miR-23a-3p, miR-27a-3p, miR-30e-5p, miR-92b-3p, miR-125b-5p, miR-199a-3p, miR-223-3p and miR-424-5p. No common miRNA was found in brain tissues and EVs derived from CSF, potentially because of the limited number of upregulated miRNAs in CSF studies. Seventeen downregulated miRNAs were common in EVs and brain tissues (Figure 1F). Among all the common miRNAs, 12 miRNAs (miR-9-5p, miR-95, miR-127-3p, miR-127-5p, miR-129-5p, miR-136-5p, miR-138-5p, miR-329, miR-410, miR-433, miR-598 and miR-769-5p) were common in brain tissues and EVs of CSF only. Two miRNAs (miR-132-3p, miR-139-5p) were common in brain tissues and EVs of plasma and CSF. miR-124-3p was common in brain tissues and EVs from serum and CSF. miR-182-5p was common in brain tissues and EVs from serum. miR-146a-5p was common in brain tissues and EVs from whole blood. In total, 15 out of 17 downregulated miRNAs were common in brain tissues and EVs from CSF, consistent with the expectation that the CSF should share more common miRNAs with brain tissues than with other body fluids.

In order to know whether the diagnostic miRNAs from EVs share the common miRNAs with brain tissues, an overlap of all of the dysregulated miRNAs from EVs, brain tissues and biomarkers was drawn. As shown in Figure 1G, only miR-132-3p (AUC = 0.77) and miR-125b-5p (combined with other miRNAs, AUC = 0.83∼0.92) were common in EVs and brain tissues. miR-132-3p was downregulated in AD brains [56,84] and EVs from plasma [79] and CSF [49]. miR-125b-5p was upregulated in brains [36,48,50] and EVs from CSF of AD patients compared to HC [49,66]. We did not find miR-135a-5p dysregulated in the brain of AD patients, but it has the best performance in single-miRNA biomarkers (Table 5). In addition, another 31 dysregulated miRNAs were common in EVs and brain tissues (Figure 1G). Some of them have the same alteration tendency, such as miR-199a-3p, which was upregulated in AD brains [80] and EVs from plasma [87]. However, some of them have the opposite tendency in different types of samples, such as miR-127-3p, which was downregulated in AD brains [80] and EVs from CSF, but upregulated in EVs from serum [49]. miR-146a-5p was upregulated in brains and EVs from plasma of AD patients compared to HC [50,70,73,74,83,87]. However, some studies reported that miR-146a-5p was downregulated in brains and EVs from whole blood [31,83].

Finally, we compared five therapeutic miRNAs in Table 6 to the dysregulated miRNAs in EVs and brain tissues of AD patients in Figure 1H. Only miR-9-5p was downregulated in EVs and brain tissues of AD patients and used as a therapeutic miRNA. miR-16-5p was downregulated in AD EVs only. miR-29b and miR-124 were downregulated in brain tissues only. miR-206-3p was not found to be dysregulated in either EVs or brain tissues. The other miRNAs that were dysregulated in EVs and brains of AD patients might be therapeutic targets in future studies. miR-9-5p [42,43], miR-16-5p [101], miR-29b [47], miR-124 [59] and miR-206-3p [105] were related to the generation of Aβ. Delivering miR-9-5p, miR-16-5p, miR-29b and miR-124 and suppressing miR-206-3p have therapeutic effects (Table 6).

In summary, we compared the dysregulated miRNAs in EVs to those in brain tissues of AD patients, aiming to provide a comprehensive view of the relevant miRNAs in EVs of AD. By comparing Figure 1A,D, miR-125b-5p (miR-125) is upregulated in several different brain tissues of AD (Figure 1A) and EVs of AD (Figure 1D), and miR-132-3p (miR-132) is downregulated in several different brain tissues of AD (Figure 1A) and EVs of AD (Figure 1D). These results suggest that miR-125-5p and miR-132-3p are good markers for the diagnosis of AD based on EV miRNAs because they carry the miRNA dereglation patterns of brain tissues. We also found that miR-9-5p was normally downregulated in both EVs and brain tissues of AD patients and was tested as a therapy for AD in both mice and human cell models of AD. Therefore, miR-9-5p is a promising candidate for designing potential therapies of AD.

## Figures and Tables

**Figure 1 cells-12-01378-f001:**
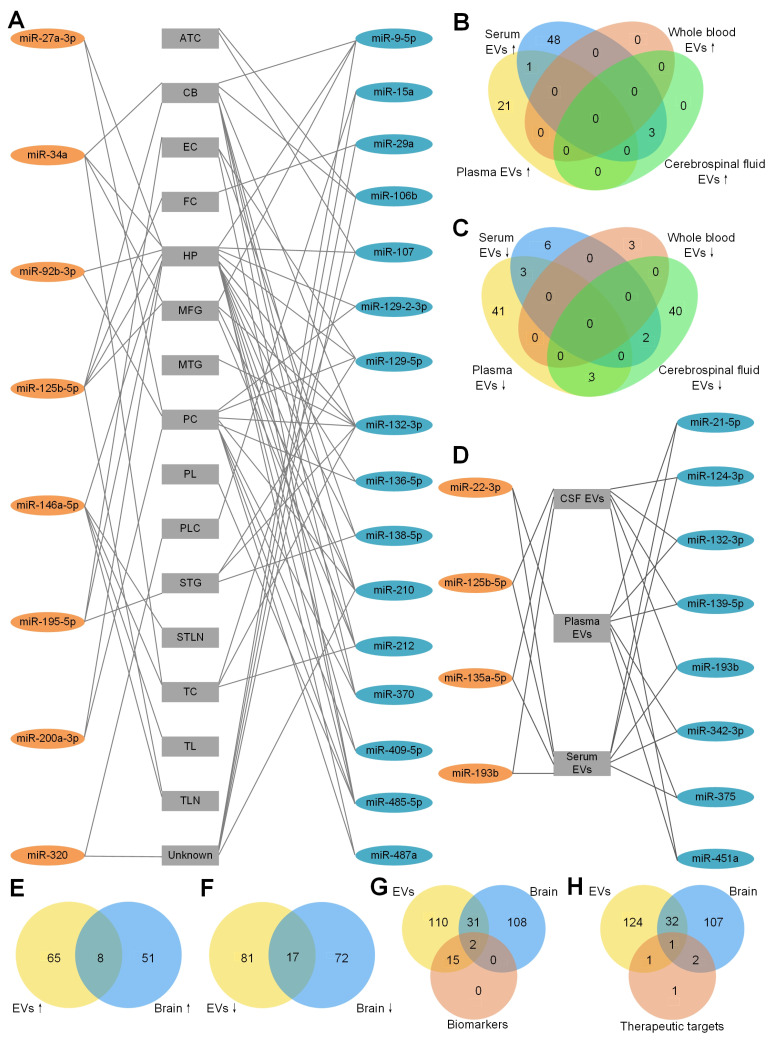
Dysregulated miRNAs in different brain regions of AD patients. The detailed legend is given on the next page. (**A**) Dysregulated miRNAs in different brain regions of AD patients. A line between a miRNA and a brain region means the miRNA is dysregulated in the brain region. These miRNAs were reported by at least 2 studies. 8 miRNAs in orange ellipses of the left column were upregulated in AD patients compared to HC. 16 miRNAs in blue ellipses of the right column were downregulated in AD patients compared to HC. Abbreviations of the different brain regions in the central column: ATC: anterior temporal cortex; CB: cerebellum; CSF: cerebrospinal fluid; EC: entorhinal cortex; EVs: extracellular vesicles; FC: frontal cortex; HP: hippocampus; MFG: medial frontal gyrus; MTG: middle temporal gyrus; PC: prefrontal cortex; PL: parietal lobe; PLC: parietal lobe cortex; STG: superior temporal gyrus; STLN: superior temporal lobe neocortex; TC: temporal cortex; TL: temporal lobe; TLN: temporal lobe neocortex; and Unknown: unknown brain region. (**B**) Upregulated miRNAs of EVs derived from plasma, serum, whole blood and cerebrospinal fluid (CSF). There is no upregulated miRNA in exosomes derived from whole blood. Among the upregulated miRNAs from serum EVs, only miR-22-3p is common in serum EVs and plasma EVs and 3 miRNAs (miR-125b-5p, miR-135a-5p, miR-193b) are common in serum EVs and CSF EVs. (**C**) Downregulated miRNAs of EVs derived from plasma, serum, whole blood and CSF. There is no common miRNA in plasma EVs, serum EVs and whole-blood EVs. Among the upregulated miRNAs from serum EVs, 3 miRNAs (miR-21-5p, miR-342-3p, miR-375) are shared with plasma EVs and 2 miRNAs (miR-124-3p, miR-193b) are shared with CSF EVs. There are 3 (miR-132-3p, miR-139-5p, miR-451a) downregulated miRNAs in both plasma EVs and CSF EVs. (**D**) Dysregulated miRNAs in EVs of different body fluids of AD patients. These miRNAs are dysreguated in EVs from 2 different body fluids. 4 miRNAs in orange ellipse of the left column were upregulated in AD patients compared to HC. 8 miRNAs in blue ellipse of the right column were downregulated in AD patients compared to HC. (**E**) EVs and brain tissue have 8 upregulated miRNAs in common. (**F**) EVs and brain tissue have 17 downregulated miRNAs in common. (**G**) MiRNAs as AD biomarkers and all of the dysregulated miRNAs from EVs and brain tissue. Among all 17 miRNAs in EVs that could be an AD biomarker, only miR-125b-5p and miR-132-3p are dysregulated in EVs and brain tissue. Moreover, another 31 common miRNAs are dysregulated in EVs and brain tissue. (**H**) MiRNAs as AD therapeutic targets and all of the dysregulated miRNAs from EVs and brain tissue. For the 5 therapeutic miRNAs, miR-16-5p is dysregulated in EVs only. miR-29b and miR-124 are dysregulated in brain only. Only miR-9-5p was dysregulated in EVs and brain of AD patients and was tested as a therapeutic target of AD. miR-206-3p was not found to be dysregulated in either EVs or brain tissue.

**Table 3 cells-12-01378-t003:** Dysregulated EV miRNAs in blood of AD patients.

Sample (AD/Control)	Upregulated miRNAs	Downregulated miRNAs	Reference
**Plasma EVs**			
21 (10/11)	/	let-7i-5p, miR-21-5p, miR-23a-3p, miR-126-3p, miR-151a-3p, miR-451a	[86]
70 (35/35)	miR-138-5p, miR-548at-5p, miR-659-5p, miR-5001-3p	miR-23b-3p, miR-24-3p, miR-29b-3p, miR-125b-5p, miR-139-5p, miR-141-3p, miR-150-5p, miR-152-3p, miR-185-5p, miR-338-3p, miR-342-3p, miR-342-5p, miR-3065-5p, miR-3613-3p, miR-3916, miR-4772-3p	[85]
47 (31/16)	/	miR-132-3p, miR-212-3p	[79]
80 (40/40)	miR-23a-3p, miR-223-3p, miR-190a-5p	miR-100-3p	[88]
25 (5/20)	miR-21-5p, miR-22-3p, miR-23a-3p, miR-27b-3p, miR-27a-3p, miR-28-3p, miR-146a-5p, miR-151a-5p, miR-199a-3p, miR-369-5p, miR-378i, miR-379-5p, miR-382-5p, miR-423-5p, miR-576-5p	let-7b-3p, let-7b-5p, let-7e-5p, miR-17-5p, miR-19b-3p, miR-20a-5p, miR-122-5p, miR-125a-5p, miR-183-5p, miR-191-3p, miR-193b-5p, miR-197-3p, miR-197-5p, miR-204-5p, miR-320e, miR-375, miR-483-3p, miR-1468-5p, miR-3173-5p, miR-3591-3p, miR-4659a-3p, miR-6749-3p	[87]
51 (23/28)	let-7e-5p	/	[69]
**Serum EVs**			
115 (53/62)	miR-22-5p, miR-30c-2-3p, miR-34b-3p, miR-34b-5p, miR-34c-5p, miR-125a-3p, miR-125b-1-3p, miR-127-3p, miR-135a-5p, miR-184, miR-219-2-3p, miR-671-3p, miR-873-3p, miR-887, miR-1285-3p, miR-1307-5p, miR-3176	miR-21-5p, miR-182-5p, miR-375	[49]
59 (31/28)	miR-15a-5p, miR-18b-5p, miR-20a-5p, miR-30e-5p, miR-93-5p, miR-101-3p, miR-106a-5p, miR-106b-5p, miR-143-3p, miR-335-5p, miR-361-5p, miR-424-5p, miR-582-5p, miR-3065-5p	miR-15b-3p, miR-342-3p and miR-1306-5p	[90]
436 (208/228)	miR-135a-5p, miR-384	miR-193b	[91]
333 (263/60)	miR-193b	/	[92]
16 (8/8)	miR-22-3p, miR-22-5p, miR-92b-3p, miR-125b-5p, miR-193a-5p, miR-194-5p, miR-320a-3p, miR-320b, miR-320c, miR-320d, miR-320e, miR-375-3p, miR-378a-3p, miR-378c, miR-378d, miR-451a, miR-1180-3p, miR-2110, miR-3615, miR-4508	miR-30b-5p, miR-124-3p, miR-144-5p, miR-223-3p	[93]
80 (40/40)	miR-22-3p, miR-378a-3p	miR-30b-5p	
540 (510/30)	miR-135a-5p	/	[94]
144 (74/74)	/	miR-193b	[58]
**Whole blood**			
32 (13/19)	/	miR-146a-5p	[31]
28 (9/19)	/	let-7g, miR-142-3p, miR-146a-5p	[31]

**Table 4 cells-12-01378-t004:** Dysregulated EV miRNAs in CSF of AD patients.

Sample (AD/Control)	Upregulated miRNAs	Downregulated miRNAs	Reference
29 (17EOAD/12)	miR-125b-5p	miR-16-5p, miR-451a, miR-605-5p	[36]
25 (13LOAD/12)	miR-125b-5p	miR-451a, miR-605-5p	
127 (62/65)	/	miR-9-3p, miR-9-5p, miR-10a-5p, miR-33b-5p, miR-95, miR-101-5p, miR-124-3p, miR-127-3p, miR-127-5p, miR-129-5p, miR-132-3p, miR-134, miR-136-3p, miR-136-5p, miR-138-5p, miR-139-5p, miR-181a-3p, miR-181a-5p, miR-181b-5p, miR-181d, miR-184, miR-199b-5p, miR-218-5p, miR-323a-3p, miR-326, miR-329, miR-377-5p, miR-381, miR-410, miR-431-3p, miR-433, miR-488-3p, miR-495, miR-598, miR-708-3p, miR-708-5p, miR-760, miR-769-5p, miR-873-5p, miR-874, miR-3200-3p	[49]
33 (27/6)	miR-193b	/	[92]
39 (32/7)	miR-135a-5p	/	[94]
14 (7/7)	/	miR-193b	[58]

**Table 5 cells-12-01378-t005:** Dysregulated miRNAs as biomarkers of AD.

Sample (AD/Control)	Sample Source	MiRNAs	AUC	Accuracy (%)	Reference
29 (17EOAD/12)	Exosomes in CSF	miR-451a	0.95	/	[36]
25 (13LOAD/12)			0.85	/	
70 (35/35)	Exosomes in plasma	miR-23b-3p, miR-24-3p, miR-125b-5p, miR-141-3p, miR-152-3p, miR-342-3p, miR-342-5p	0.83-0.92	83-89	[85]
47 (31/16)	Exosomes in plasma	miR-132-3p	0.77	/	[79]
		miR-212-3p	0.84	/	
51 (23/28)	EVs in plasma	let-7e-5p	0.92	/	[69]
436 (208/228)	Exosomes in serum	miR-135a-5p	0.98	/	[91]
		miR-193b	0.80	/	
		miR-384	0.87	/	
		miR-135a-5p, miR-193b	0.99	/	
		miR-135a-5p, miR-384	0.98	/	
		miR-193b, miR-384	0.90	/	
		miR-135a-5p, miR-193b, miR-384	0.995	/	
80 (40/40)	Exosomes in serum	miR-22-3p	0.64	/	[93]
		miR-30b-5p	0.67	/	
		miR-378a-3p	0.72	/	
		miR-22-3p, miR-30b-5p, miR-378a-3p	0.88	/	

**Table 6 cells-12-01378-t006:** MiRNAs as potential targets of AD treatment.

MicroRNA	Mechanism of Treatment	AD Model	Reference
miR-9-5p	Has the potential ability of neuroprotection by decreasing the expression of CAMKK2 and AMPK.	Mouse and Human Cells	[100]
	Might improve the differentiation of neural stem cells into neurons through preventing the activation of the Notch signaling pathway.	Cells	[99]
miR-16-5p	Brain delivery of miR-16 mimics induces a reduction in the AD-related genes APP, BACE1 and Tau protein.	Mice	[101]
	Upregulated miRNA decreased the apoptosis of primary hippocampal neurons.	Cells	[102]
miR-29b	miR-29b-containing exosome delivery proved the potential therapeutic effects of Aβ by reducing the pathological effects of Aβ. The target genes of miR-29b were BACE1 and BIM.	Rat	[104]
miR-124	Combined with rutin, inhibits the expression of BACE1 and APP.	Mice	[103]
miR-206-3p	Improved synaptic plasticity and cognitive deficits and attenuated Aβ accumulation by inhibiting the expression of miR-206-3p.	Mice	[105]

## Data Availability

Not applicable.

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
