# Peer review of "MicroRNAs in Extracellular Vesicles of Alzheimer’s Disease"

_cells, 2023, doi:10.3390/cells12101378_

Round 1
Reviewer 1 Report
This is an important, complete, and well organized review of the microRNAs that have been found/studied in connection with Alzheimer's disease.
Here are the minor revisions I would like to suggest:
1) Please add the Braak stage information in Table 2.
2) Please move the paragraph describing the Braak stages (lines 150-156) after the paragraph introducing the studies shown in Table 2 (lines 157-163)
3) Please introduce the paragraph describing the Braak stages by a statement like "In these studies, the Braak stage was often specified and we have included this information in Table 2 when available".
4) Please review the English writing as some sentences are either incorrect or missing some words.
Author Response
Reviewer reports:
Independent Review Report, Reviewer 1:
Comments 1.0 This is an important, complete, and well organized review of the microRNAs that have been found/studied in connection with Alzheimer's disease.
Responses 1.0
Thank you.
Here are the minor revisions I would like to suggest:
Comments 1.1 Please add the Braak stage information in Table 2.
Responses 1.1
Thanks. We have added the Braak stage information in Table 2.
Comments 1.2 Please move the paragraph describing the Braak stages (lines 150-156) after the paragraph introducing the studies shown in Table 2 (lines 157-163)
Responses 1.2
This is a good comment. We have moved the paragraph describing the Braak stages (lines 150 - 156) after the paragraph introducing the studies shown in Table 2 (page 5, line 164 - 172).
Comments 1.3 Please introduce the paragraph describing the Braak stages by a statement like "In these studies, the Braak stage was often specified and we have included this information in Table 2 when available".
Responses 1.3
Thank you. We have added a sentence “The Braak stage was often used in AD studies and we summarized the information about Braak stage in Table 2 when available” on page 5, lines 170 - 172.
Comments 1.4 Please review the English writing as some sentences are either incorrect or missing some words.
Responses 1.4
Thank you for your comment. We carefully revised the manuscript and improved the language and writing. As also listed in Responses 3.8, we corrected many incorrect language problems.
Reviewer 2 Report
In this work, the authors highlighted that miR-125b-5p (miR-125) is an upregulated miRNA in several AD (Alzheimer's disease) brain tissues and EVs (extracellular vesicles) of AD, while miR-132-3p (miR- 132) is a downregulated miRNA in several AD brain tissues and EVs of AD. These results suggest that miR-125-5p and miR-132-3p are good markers for miRNA-based diagnosis of AD in EVs. In addition, miR-9-5p have been deregulated in EVs and different brain tissues of AD patients and tested as potential therapies for AD in human and mouse cell models, suggesting that miR-9-5p could be used to design new AD therapies. i think the data on miR-125-5p and miR-132-3p is well exposed and interesting in that it shows that the data on miRNAs that we can obtain from EVs is similar to that which can be obtained from the brain tissue of subjects with AD. This surely can play an important role in the diagnosis of AD. As for the data obtained for miR-132-3p, i think that the terpeutic aspect should be avoided.
Author Response
Independent Review Report, Reviewer 2:
Comments 2.0 In this work, the authors highlighted that miR-125b-5p (miR-125) is an upregulated miRNA in several AD (Alzheimer's disease) brain tissues and EVs (extracellular vesicles) of AD, while miR-132-3p (miR- 132) is a downregulated miRNA in several AD brain tissues and EVs of AD. These results suggest that miR-125-5p and miR-132-3p are good markers for miRNA-based diagnosis of AD in EVs. In addition, miR-9-5p have been deregulated in EVs and different brain tissues of AD patients and tested as potential therapies for AD in human and mouse cell models, suggesting that miR-9-5p could be used to design new AD therapies. I think the data on miR-125-5p and miR-132-3p is well exposed and interesting in that it shows that the data on miRNAs that we can obtain from EVs is similar to that which can be obtained from the brain tissue of subjects with AD. This surely can play an important role in the diagnosis of AD. As for the data obtained for miR-132-3p, I think that the terpeutic aspect should be avoided.
Responses 2.0
Thank you very much.
For miR-132-3p, we did not mention its potential as therapeutic target. As listed in Table 6, five miRNAs, i.e., miR-9-5p, miR-16-5p, miR-29b, miR-124 and miR-206-3p, were potential targets of AD treatment based on existing results.
Reviewer 3 Report
This is an interesting review on epigenetic mechanisms related to miRNAs that underlie pathologic mechanisms in Alzheimer’s disease (AD).
Interesting aspects:
- The review brings into focus miRNAs in extracellular vesicles, as a mechanism controlling intercellular signalling in the brain and blood (plasma, serum, whole blood) of AD patients.
- The authors review a large amount of clinical data supporting the involvement of specific miRNAs in AD, in addition to the data generated in preclinical models.
There are several issues that should be revised by the authors:
- To define very well the message that authors intend to transmit in each paragraph, and to reword accordingly to make the message clear for the readers. Please, revise the logic of the argumentation.
- To avoid repeating in the text the information contained in the tables. In the text, the most interesting studies could be presented in more detail or from another perspective. It was interesting that miRNAs enumerated in Table 2 are discussed from the perspective of the Braak stage.
- To comment on the functional role of those particular miRNAs that are presented in more detail in the text, including their target genes.
- To comment more on the methods used for EV isolation which might be the cause of discrepancies between the findings of different studies (mentioned by the authors)
- Chapter 7. Discussion is in generally a continuation of Chapter 6. Therefore, the information provided in Chapter 7 should be inserted in Chapter 6.
- To provide a “Future perspectives” section instead of Discussion.
- To revise extensively the English language because the text is quite difficult to read (at least for me). The revision of the paper by an English language native is necessary.
Find attached a commented version of the paper with more details than provided above. Besides comments I marked in yellow the sentences that should be revised and some English language corrections.
At the second revision, we may further improve the article, which is really interesting.

Author Response
Independent Review Report, Reviewer 3:
Comments 3.0 This is an interesting review on epigenetic mechanisms related to miRNAs that underlie pathologic mechanisms in Alzheimer’s disease (AD).
Interesting aspects:
-The review brings into focus miRNAs in extracellular vesicles, as a mechanism controlling intercellular signalling in the brain and blood (plasma, serum, whole blood) of AD patients.
-The authors review a large amount of clinical data supporting the involvement of specific miRNAs in AD, in addition to the data generated in preclinical models.
Responses 3.0
Thank you.
There are several issues that should be revised by the authors:
Comments 3.1 To define very well the message that authors intend to transmit in each paragraph, and to reword accordingly to make the message clear for the readers. Please, revise the logic of the argumentation.
Responses 3.1
We had corrected many language errors in the original manuscript, as given in Responses 3.8.
Comments 3.2 To avoid repeating in the text the information contained in the tables. In the text, the most interesting studies could be presented in more detail or from another perspective. It was interesting that miRNAs enumerated in Table 2 are discussed from the perspective of the Braak stage.
Responses 3.2
Thank you. As suggested, by the first reviewer too, we also added Braak stage of these studies when available.
Comments 3.3 To comment on the functional role of those particular miRNAs that are presented in more detail in the text, including their target genes.
Responses 3.3
As listed in Table 1, we included the targets of miRNAs when they had been reported in these studies. In Section 2, “Relevant miRNAs in AD”, lines 97 - 155, the target genes of miRNAs were also mentioned.
Comments 3.4 To comment more on the methods used for EV isolation which might be the cause of discrepancies between the findings of different studies (mentioned by the authors)
Responses 3.4
We carefully revised the related parts on page 8, lines 268 – 271. In [95], all exosomes in serum of peripheral blood were analyzed and in [96], ABCA1-labeled exosomes were captured and analyzed.
Comments 3.5 Chapter 7. Discussion is in generally a continuation of Chapter 6. Therefore, the information provided in Chapter 7 should be inserted in Chapter 6.
Responses 3.5
Section 6, “Application of miRNAs in AD treatment”, is intended to give a review of miRNAs that have been tested as the therapeutic targets of AD in vitro or disease model animal experiments. Moreover, we intended to summarized and compare the dysregulated miRNAs in different brain regions and EVs of different body fluids in Section 7, Discussion. Therefore, we think that it is suitable to keep Discussion as a separate section.
Comments 3.6 To provide a “Future perspectives” section instead of Discussion.
Responses 3.6
We included at the end of Discussion a paragraph to summarize the whole manuscript and to provide some perspectives.
Comments 3.7 To revise extensively the English language because the text is quite difficult to read (at least for me). The revision of the paper by an English language native is necessary.
Responses 3.7
We are very sorry for this. We had carefully corrected the language errors in the revised manuscript, as listed in Responses 3.8.
Comments 3.8 Find attached a commented version of the paper with more details than provided above. Besides comments I marked in yellow the sentences that should be revised and some English language corrections.
At the second revision, we may further improve the article, which is really interesting.
Responses 3.8
Thank you very much for your kind comments, which are very helpful for us to improve the manuscript. All the comments that made in the attach file have been revised as following.
On page 1, line 1, we have changed “disease with memory, language and thinking dysfunction” to “disease with dysfunction of memory, language and thinking.”
On page 1, line 6, we have changed “miRNAs which involve in” to “miRNAs which are involved in”.
On page 1, line 13, we have changed “and also tested as…” to “and had also been tested as…”. We want to say miR-9-5p had also been tested as potential therapies of AD in mice and human cell models.
On page 1, line 13 and line 14, we have changed “therapies of AD” to “therapies for AD”.
On page 1, line 18, we have changed “disease with memory, language and thinking dysfunction” to “disease with dysfunction of memory, language and thinking.”
On page 1, line 19, we have changed “World Health Organization” to “the data of World Health Organization”.
On page 1, line 27, we have changed “the Aβ” to “Aβ”.
On page 1, line 28, we changed “cells of brain” to “brain cells”.
On page 1, line 29, we changed “Tau protein involves in the microtubule formation that is related to the cell…” to “are involved in microtubule formation that is related to cell…”
On page 1, line 34, we have changed “Aβ and Tau acting collaborative function in…” to “that Aβ and Tau are acting collaboratively in…”.
On page 1, line 36 - 38, we added a sentence about the limitation of these therapies as suggested, “A review summarizing treatment strategies of AD indicated that therapies of Aβ did not get the results as expected in clinical trials [11]. But therapies of Tau are in the early stage of development and has a great therapeutic potential in the future [11].”
On page 2, line 39, we have changed “the two most popular pathologies of AD” to “the amyloid cascade and Tau hypotheses of AD”.
On page 2, line 41, we have changed “late onset Alzheimer’s disease” to “late onset AD”.
On page 2, line 42, we have changed “ApoE” to “the ApoE”.
On page 2, line 43, we have changed “a variant of the APOE gene production” to “a variant protein product of the APOE gene”.
On page 2, line 46, we have changed “proceeding neuronal dysfunction” to “promoting neuronal dysfunction”. The suggested “inflicting” is incorrect in this case.
On page 2, line 47, we have changed “immune network” to “the immune network”.
On page 2, line 48, we have changed “summarized the role of…” to “Gratuze et al. [17] summarized the role…”
On page 2, line 50, we have changed “immune response” to “the deregulated immune response”.
On page 2, line 57, we did not change “Aβ deposition induced the microglial activation” to “Aβ deposition induced by microglial activation” as you suggested, because the microglial activation is the result of Aβ deposition not the cause.
On page 2, line 56 - 57, we have changed “cerebrospinal fluid (CSF) showed that increased Aβ production of sleep-deprived” to “cerebrospinal fluid (CSF) showed increased Aβ production in sleep-deprived”.
On page 2, line 59 - 60, we revised the second sentence of this paragraph.
On page 2, line 59 and line 61, we eliminated “the” before “miRNAs” and “pri-miRNA”.
On page 2, line 63 - 64, we have changed “The pri-miRNA is transported to the cytoplasm and cleaved the loop” to “Then pri-miRNA is transported to the cytoplasm and the loop is cleaved”.
On page 2, line 63 - 66, we have changed the original sentences to “Then pre-miRNA is transported to the cytoplasm and the loop is cleaved to produce a double-strand RNA called miRNA:miRNA* duplex. One strand of the miRNA:miRNA* duplex is loaded with AGO protein to form the silencing complex. Meanwhile the other strand is discarded”.
On page 2, line 67 - 68, we have changed “MiRNAs usually suppress the expression of their targets, by complementary binding to the target mRNAs.” to “MiRNAs usually suppress the expression of their targets by complementary binding to the target mRNAs.”
On page 2, line 71, we added an introductory sentence “Extracellular vesicles (EVs) are membrane-enclosed vesicles secreted by cells.”
On page 2, line 75, we have changed “released from the healthy cells” to “healthy living cells”.
On page 2, line 76, we have changed “by programmed cell death” to “during programmed cell death”.
On page 2, line 76 - 79, we have changed “A study of distinct RNA in extracellular vesicles from three different kinds of cell lines (BV-2, HMC-1 and TF-1) indicated that there was a little or no RNA in microvesicles except for that collected from TF-1 cells” to “A study of distinct RNA in extracellular vesicles from three different kinds of cell lines including a human erythroleukemia cell line (TF-1), a human mast cell line (HMC-1), and a mouse microglia cell line (BV-2), indicated that there was a little or no RNA in microvesicles except for that collected from TF-1 cells”.
On page 2, line 80 - 82, we have changed “Meanwhile, Budnik et al. thought apoptotic bodies might not involve in transcellular communication in the nervous system, since they were engulfed by phagocytic cells rapidly as being released” to “Meanwhile, Budnik et al. [37] thought that apoptotic bodies might not be involved in transcellular communication in the nervous system, since they were engulfed by phagocytic cells rapidly as being released”.
On page 2, line 82 - 83, we have changed “Among the three types of EVs, exosomes involve in cell-to-cell communication in central nervous system” to “Only exosomes are involved in cell-to-cell communication in the central nervous system”.
On page 2, line 83 - 84, we have changed “Exosomes from neurons regulate integrity of brain vascular system” to “Exosomes from neurons regulate the integrity of the brain vascular system”.
On page 2, line 87 - 88, we have changed “MiRNAs of exosomes remain similarities with its parent cells in cancer and proteins of EVs consist of proteins of membrane and lumen of EVs.” to “MiRNAs of exosomes keep similarities with the parent cells in cancer and proteins of EVs consist of membrane proteins and lumen of EVs”.
On page 2, line 89, we have changed “pathological changes of miRNAs” to “pathological miRNAs”.
On page 2, line 90 - 91, we have changed “Indeed, it has been reported that miRNAs in exosomes alter in AD patients.” to “Indeed, it has been reported that miRNAs in exosomes are dysregulated in AD patients when compared to healthy people.”
On page 3, line 97, we changed the related part to “, as summarized in Table 1”.
On page 3, line 98, we have changed “abundant researches” to “many researches".
On page 4, line100, we have changed “amyloid precursor protein (APP)” to “the amyloid precursor protein (APP)”
On page 4, line101, we have changed “resulting that” to “resulting in”.
On page 4, line102 - 103, we have changed “protective factor of AD by inhibiting accumulation of Aβ” to “protective factor in AD by inhibiting the accumulation of Aβ”.
On page 4, line 108, we have changed “temporal lobe neocortex of AD brain” to “the temporal lobe neocortex of AD brain”.
On page 4, line 109 - 110, we added a sentence about the functional connection between the targets of miR-9-5p. “All these targets of miR-9-5p are related to the Aβ accumulation in AD.”
On page 4, line 111, we have changed “through targeting UBE4B” to “by targeting UBE4B”.
On page 4, line 112, we have changed “AD patient” to “AD patients”.
On page 4, line 117, we have changed “cell apoptosis” to “apoptosis”.
On page 4, line 117, we have changed “Zhang et al. thought apoptosis might associate with …” to “Zhang et al. [78] reported that apoptosis might be associated with …”.
On page 4, line 123 - 124, we added a sentence about the connection between findings. “Therefore miR-9-5p is also involved in AD development through other ways, such as apoptosis and oxidative stress.”
On page 4, line 125, we have changed “AD pathology too” to “AD pathology too. As listed in Table 1…”.
On page 4, line 127, we have changed “the Aβ” to “Aβ”.
On page 4, line 128, 129, 130, 133, 135, we have changed “(Table 1)” to “(as listed in Table 1)”.
On page 4, line 130, 136, 140, 142, 143, we have changed “(Table 1)” to “(see Table 1).
On page 4, line 130, we have changed “the inhibitors of ADAM10” to “inhibitors of ADAM10”
On page 4, line 134 - 135, we have changed “the inhibitors of BACE1 too (Table 1).” to “also BACE1 inhibitors (as listed in Table 1).”.
On page 4, line 136, we have changed “the Tau” to “Tau”.
On page 4, line 137 - 142, we have changed “For example, miR-15a might be a potential regulator of ERK1, a candidate of kinase in Tau phosphorylation, and is involved in the AD development [48]. miR-26b was up-regulated in AD and might induce cell apoptosis, aberrant cell cycle entry and increasing Tau-phosphorylation [50] (Table 1). miR-125b-5p was involved in the Tau hyperphosphorylation by up-regulating the expression of Erk1/2, an activator of kinases cdk5 and decreasing phosphatases production including DUSP6 and PPP1CA [56] (Table 1).”.to “For example, miR-15a might be a potential regulator of ERK1[54]. miR-125b-5p up-regulated the expression of Erk1/2, an activator of kinases cdk5, and involved in Tau phosphorylation in AD development [62]. miR-125b-5p also decreases production of phosphatases, including DUSP6 and PPP1CA [62] (see Table 1). miR-26b was up-regulated in AD and might induce cell apoptosis, aberrant cell cycle entry and increasing Tau-phosphorylation [56] (see Table 1).”
On page 4, line 147 - 148, we have changed “let-7e-5p also triggered the increasing of pro-inflammatory cytokine, such as interleukin 6 and inflammation in microglial” to “let-7e-5p also triggered the increase of pro-inflammatory cytokines (such as interleukin 6) and inflammation in microglial”.
On page 4, line 151, we have changed “the energy metabolism” to “energy metabolism”
On page 4, line 153, we have changed “miR-212 and inhibited the apoptosis” to “miR-212 inhibited apoptosis”.
Comments: On page 5, line 156, “Please clearly devise the chapter in a section related to brain "regions (Table 2) and the correlation with the Braak stage.” –from reviewer
Responses:
We added Braak stages of different studies in Table 2, when available. We also added one paragraph about Braak stage on Page 5, lines 165-173.
On page 5, line 157 - 159, as you suggested an introductory sentence is needed in reference to Table 2, we have changed the sentence to “In order to know whether there are common dysregulated miRNAs in the EVs and brain tissues of AD patients, we summarized dysregulated miRNAs in different brain regions of AD patients in Table 2.”
On page 5, line 164 - 172, we have put the paragraph described the Braak stages in here.
Comments: On page 5, line 174, “I think that you should not mention here this small study”—from reviewer.
Responses: We kept this sentence because it reported down-regulation of miR-29a in AD patients which is consistent with results in [62].
On page 5, line 161, 173, 176, 178, 182, 184, 186, 193, 198, 201; and on page 7, line 208, 209, 210, 215, 217, 219, we have changed “(Table 2)” to “(see Table 2)” or “(as listed in Table 2”.
On page 5, line 175, we have changed “(the Braak V and VI)” to “(Braak V and VI)”.
On page 5, line 176, we have changed “(the Braak I and II)” to “(Braak I and II)”
On page 5, line 179, we have changed “mild (the Braak III) and severe AD (the Braak VI)” to “mild (Braak III) and severe AD (Braak VI)”.
On page 5, line 180-181, we have changed “Braak I was regards as the normal control, and the Braak III - VI were regards as AD” to “Braak I was regarded as the normal control, and the Braak III - VI were regarded as AD”.
On page 5, line 181, 183, 198, 202, and on page 7, line 207, 209, 218, 233, 235, we have changed “AD brain” to “AD brains”.
On page 5, line 187, we have changed “compare to HC” to “as compared to HC”.
On page 5, line 188, we have changed “the other pathological features” to “other pathological features”
On page 5, line 190, we have changed “20-individuals study” to “20-individual study”.
Comments: On page 5, line 201, “Make a more clear connection between the Braak stage and miRNAs deregulation.” –from reviewer.
Responses: We had included the Braak stages of the dysregulated miRNAs in Table 2.
On page 7, line 212, we have changed “the patients description” to “the description of patients”.
Comments: On Page 7, “Please insist on the function of miRNAs, beyond their ennumeration”.
Responses: This section is for Dysregulated miRNAs in brain tissues of AD patients. Therefore, we summarized dysregulated miRNAs in Table 2 and mentioned these results in this section.
On page 7, line 221 - 222, we have changed “The expression level of miRNAs not only altered between AD patients and normal people, but also changed from different stages of AD development.” to “The expression levels of miRNAs altered not only at different stages of AD development but also in different brain regions.”
On page 7, line 223, we have changed “(the Braak III and IV)” to “(Braak III and IV)”.
On page 7, line 224, we have changed “(the Braak VI)” to “(Braak VI)”.
On page 7, line 224, we have changed “(the Braak I and II)” to “(Braak I and II)”.
On page 7, line 234, we have changed “and same result did not found in the other neurogenic disease” to “but not in other neurogenic disease”.
On page 7, line 240, we have changed “miRNAs of EVs in blood” to “miRNAs in blood EVs”.
On page 7, line 241, we eliminated the sentences “The deregulated miRNAs in EVs from different body fluids will be summarized as follows.” since you suggested that “too many comments on what you intend to do”.
On page 7, line 242, we have changed “including plasma” to “comprising plasma”.
On page 7, line 243, we have changed “easy-take” to “easy-to-collect”.
On page 7, line 247 - 249, we have changed “6 miRNAs were down-regulated in EVs derived from plasma of AD patients compared with HC [84]. But no miRNA was found statistically up-regulated in the same research [84] (Table 3).” to “In another study, 6 miRNAs were down-regulated in EVs derived from plasma of AD patients compared with HC, but no miRNAs was statistically up-regulated in the same research [90] (see Table 3).”
Comments: On page 7, line 249, “It is interesting to make some comments on the biological pathway in which these miRNAs are involved and how they related to AD. Mention that these miRNAs regulate systemic processes.” –from reviewer
Responses: The purpose of this section is to list the dysregulated miRNAs in EVs of different body fluids of AD patients. It is beyond our capacities and scope of the manuscript to comment on the biological pathways for every dysregulated miRNAs. We listed the relevant miRNAs in AD in Table 1.
On page 7, line 254, we have changed “EVs derived from plasma” to “EVs collected from plasma”.
On page 7, line 255 - 257, we have changed “the neural EVs which is not mentioned” to “neural EVs which are not mentioned”.
Comments: On page 8, line 255, “They were not mentioned because it was not assessed, or they were not found?” –from reviewer
Responses: These studies did not clearly mention that they had separated or assessed the neuron-derived EVs. The EVs in blood should include some EVs released from neurons. Otherwise, we cannot find miRNAs with the same changes in both brain tissues and body fluid EVs of AD patients.
Comments: On page 8, line 259, “Please, clarify the message” –from reviewer
Responses: This is an unclear statement and incorrect sentences. We rewrote these sentences as below (on Page 8, line 257-261).
Cheng et al. [93] indicated that the expression of miRNAs in EVs from peripheral blood had lower correlation with brain when compared to the EVs from brain tissues because other organs also released EVs to blood. The isolation method of brain-derived EVs in blood should be emphasized to find more AD-related miRNAs [93].
On page 8, we have changed “Table 3. Deregulated miRNAs in EVs of blood and whole blood samples of AD patients.” to “Table 3. Dysregulated EV miRNAs in blood of AD patients.”
On page 8, line 261, we removed three sentences as below because they are irrelevant to the main theme of the manuscript.
Serum is the liquid of blood remained after clotting. And because of that, serum lacks clotting factors compared with plasma. Plasma is the liquid of blood remained after anticoagulant is added to prevent clotting. Based on their different composition, it might provide a different sight of miRNAs in serum and plasma.
On page 8, line 262, 264, 266, 268, on page 9, line 273, 275, 278, 281, 282, we have changed “(Table 3)” to “(see Table 3)” or “(as listed in Table 3)”.
On page 8-9, line 269 - 272, we revised the related parts as blow.
The reason for the opposite change of miR-193b from the last 2 studies might be the different methods of exosome isolation applied. In [95], all exosomes in serum of peripheral blood were analyzed and in [96], ABCA1-labeled exosomes were captured and analyzed.
Comments: On page 9, line 272, “Please comment on the physiologic roles of these miRNAs.” –from reviewer
Responses: The purpose of this section is to list the dysregulated miRNAs in EVs of different body fluids of AD patients. This is beyond our capacities and scope of the manuscript to comment on every dysregulated miRNAs. We listed the relevant miRNAs in AD in Table 1.
On page 9, line 278, we have changed “the quantitative PCR (qPCR)” to “quantitative PCR (qPCR)”.
On page 9, line 280, we have changed “the whole blood” to “whole blood”.
Comments: On page 9, line 285, “Comment on the role of these miRNAs in connection with AD pathology.” –from reviewer
Responses: The purpose of this section is to list the dysregulated miRNAs in EVs of different body fluids of AD patients. This is beyond our capacities and scope of the manuscript to comment on every dysregulated miRNAs. Actually, we listed some of the relevant miRNAs in AD in Table 1.
On page 9, line 291 - 292, we have changed “Based on the social factors, like retirement age (65 years old), instead of any biological differences, AD is divided into the early onset Alzheimer’s disease (EOAD) and late onset Alzheimer’s disease (LOAD) [95].” to “AD is divided into the early onset Alzheimer’s disease (EOAD) and late onset Alzheimer’s disease (LOAD) [95].”
On page 9, line 294, 299, 301, 302, 305, we have changed “(Table 4)” to “(see Table 4)” or “(as listed in Table 4)”.
We changed on page 9, “Table 4. Deregulated miRNAs in EVs of CSF of AD patients.” to “Table 4. Dysregulated EV miRNAs in CSF of AD patients.”
On page 12, line 403 - 405, we compared CSF with serum/plasma studies in Figure 1D and related parts in Discussion.
On page 10, line 307, we have changed “miRNAs of EVs in other biological fluids” to “miRNAs in EVs of other biological fluids”.
On page 10, line 308 - 309, we have changed “Choi et al. found that some AD-related genes expressed in corneal fibroblasts and the corneal epithelium, such as APP, ADAM10, BACE1” to “Choi et al. [49] found that some AD-related genes, such as APP, ADAM10, and BACE1, were expressed in corneal fibroblasts and the corneal epithelium”.
On page 10, line 312, we have changed “…biomarkers for AD” to “…biomarkers for AD that could be detected in various organs”.
On page 10, line 314, we have changed “tears for AD” to “tears in AD”.
On page 10, line 315, we have changed “Application of miRNAs of EVs” to “Application of EV miRNAs”.
On page 10, line 316, we have changed “the AD biomarker” to “AD biomarker”.
On page 10, line 318, we have changed “the ROC analysis” to “ROC analysis”.
On page 10, line 319, we have changed “as shown in Table 5” to “as listed in Table 5”.
On page 10, line 322, we have changed “exosome” to “exosomes”.
On page 10, line 321, 322, 326, 330, we have changed “(Table 5)” to “(see Table 5)” or “(as listed in Table 5)”.
On page 11, line 340 - 342, we have changed “These neurons conversion from the specific patients can be used for the disease diagnosis, drug screening and disease model study” to “These neurons converted from iPSCs could be used for the disease diagnosis, drug screening and in disease model studies”.
Comments: On page 11, line 341, “I do not think that these methods are useful for diagnosis, but for testing personalized therapies ex vivo” –from reviewer.
Reponses: These are the statements from the authors of Ref [103]. Probably, because these cells are converted from the iPSCs, they carry the mutations of the patients. Therefore, some of these mutations might be used for diagnosis.
On page 11, line 344-345, we have changed “the therapeutic targets of AD in vitro or disease model animal experiments” to “therapeutic targets of AD in vitro or in animal model experiments”.
On page 11, line 345 - 346, we have changed “Osthole has the protective effects” to “Osthole, a natural coumarin derivative, has the protective effects”.
On page 11, line 344, 347, 351, 353, we have changed “(Table 6)” to “(see Table 6)” or “(as listed in Table 6)”.
On page 11, line 348 - 349, we have changed “And miR-9-5p might be the key point of it through decreasing the expression of CAMKK2 and p-AMPKα” to “Osthole decreases the expression of CAMKK2 and p-AMPKα through up-regulating miR-9-5p to achieve the therapeutic effects”.
On page 11, line 353, we have changed “the cellular AD…” to “cellular AD …”.
On page 11, line 356, we have changed “significantly increase the miR-124” to “significantly increased miR-124”.
On page 11, line 358, we have changed “…palyed in cell” to “played in cell”.
Comments: On page 12, line 370, “Chapter 7 is in fact part of Chapter 6” –from reviewer
Please read Responses 3.5.
Comments: On page 15, line 480, “A “Future perspectives” chapter is needed” –from reviewer
Please read Responses 3.6.
We found many other English errors in text and corrected them in the revised manuscript. Some revisions are listed as below.
We have changed the word “deregulated” to “dysregulated” in the whole text.
On page 1, line 6, we have changed “We summarize the deregulated miRNAs” to “We summarized the dysregulated miRNAs”.
On page 2, line 39, we have changed “three other hypothesises” to “three other hypotheses”.
On page 2, line 69, we have changed “pathological process” to “pathological processes”.
We have changed some description on page 3, Table 1, which had been mark in red.
On page 4, line 111, we have changed “The level of UBE4B” to “While the level of UBE4B”.
On page 4, line 120, we have changed “miR-9-5p was…” to “Moreover miR-9-5p was…”.
On page 4, line 144, we have changed “Some miRNAs have other contributions to AD development. As shown in Table 1” to “Some miRNAs have other contributions to AD development. As listed in Table 1”.
On page 5, line 173, we have changed “miR-29” to “miR-29a”.
On page 9, line 286, we have changed “Deregulated miRNAs of EVs in cerebrospinal fluid” to “Dysregulated miRNAs in cerebrospinal fluid EVs”.
On page 9, line 293-294, we have changed “found that mir-125b-5p was up-regulated and miR-16-5p, miR-451a, miR-605-5p and miR-125b-5p were down-regulated compared to HC” to “found that miR-125b-5p was up-regulated and miR-16-5p, miR-451a and miR-605-5p were down-regulated compared to HC”.
On page 10, line 330 – 332, we have changed “However the number of the samples and the performance of the model is much lower than the result of Yang et al.” to “However the number of the samples is smaller than that in Yang et al. [95] and the performance of the model is much worse than those in Yang et al.”
Round 2
Reviewer 1 Report
Responses from authors are satisfactory.
Author Response
Thank you again for your helpful comments.